# Making Pb Adsorption-Saturated Attapulgite with Excellent Photocatalysis Properties through a Vulcanization Reaction and Its Application for MB Wastewater Degradation

**DOI:** 10.3390/ijerph191610457

**Published:** 2022-08-22

**Authors:** Xiao Zhang, Chen Chen, Ting Cheng, Mingyue Wen, Lei Wang, Fenxu Pan

**Affiliations:** 1School of Environmental Ecology, Jiangsu City Vocational College, Nanjing 210017, China; 2School of Environmental and Chemical Engineering, Jiangsu University of Science and Technology, Zhenjiang 212100, China

**Keywords:** attapulgite, PbS, adsorption-saturated, photocatalysis, MB

## Abstract

Attapulgite (AT) is a clay mineral with rich reserves in China, and it has good adsorption activity for Pb-containing wastewater. However, as a hazardous waste, the treatment of Pb adsorption-saturated attapulgite was quite difficult. In this work, through a simple vulcanization reaction, the waste Pb adsorption-saturated attapulgite (AT@Pb) was transformed into composite materials (AT@PbS) with good photocatalytic performance. After comprehensive material characterization (including XRD, TEM, XPS, and UV-Vis), the photocatalytic degradation performance and mechanism of AT@PbS for methylene blue (MB) were investigated. The results revealed that AT@PbS was a composite material of attapulgite nanorods (500–600 nm) and nanosquare PbS particles (80–100 nm). Additionally, AT@PbS displayed good visible light absorption, improved photo-electric properties, excellent photodegradation performance for MB, and recycling stability. Moreover, the energy band range of AT@PbS was about −0.043 V to 1.367 V. The photo-generated holes and their derived hydroxyl radicals were the main active species for MB degradation. This work not only provides a new approach to construct the composite photocatalyst, but also demonstrates the possibility of the comprehensive utilization of heavy metal adsorbents for wastewater degradation.

## 1. Introduction

At present, many chemical industries, such as battery manufacturing [1], lead mining [2] and the petrochemical industry [3], discharge a large amount of lead-containing wastewater in the production process. The Pb element in wastewater will not only contaminate the ecological environment, but also pose a major threat to human health [4,5,6]. Accordingly, the treatment of this kind of wastewater is of great significance to all aspects of society [7,8]. Adsorption is one of most useful methods for treating Pb-containing wastewater because of its high removal efficiency for heavy metals, simple operation process, and low operating cost [9,10,11].

Attapulgite is one of the most abundant natural porous minerals in China, and has received much attention due to its chemical stability, environmental friendliness, and cost-effectiveness. As a crystalloid hydrous magnesium aluminium silicate mineral with a chain-layered structure (the theoretical chemical formula is Mg_5_Si_8_O_20_(OH)_2_(OH_2_)_4_·4H_2_O)**,** attapulgite has huge specific surface area, and provides adsorption and chemical activities that generate advantageous nano-effects. According to related references, attapulgite displays typical nanorod morphology and excellent Pb adsorption performance [12,13,14]. Pyrgaki et al., used raw and heat-treated attapulgite clay for Pb adsorption from aqueous solutions [15]. Bhagat et al., predicted a theoretical Pb adsorption process on attapulgite clay using the feasibility of data intelligence models [16]. Xu et al., applied an experimental density functional theory (DFT) calculation to study the adsorption selectivity of amino-modified attapulgite for Pb and Cu from solution [13].

However, strictly speaking, applying an adsorption method could only transfer Pb from wastewater to the solid adsorbent, and the subsequent treatment of waste adsorbent was also a thorny environmental problem. According to the Chinese national hazardous waste directory (2021 Edition), waste adsorbent is a hazardous waste, and its treatment cost is quite expensive, which further limits the wide application of adsorption materials, including attapulgite. Therefore, finding a comprehensive utilization method for waste adsorbence can not only relieve the environmental pressure, but also effectively reduce the treatment cost of adsorption.

In fact, Pb is not only a harmful element, but has also been used as a stable composite material in many fields. Lead sulfide (PbS) is one of the typical stable Pb compound materials. It is considered to be a good semiconductor material, and widely applied in the field of photocatalytic treatment of organic wastewater [17,18]. The photocatalysis technology is an effective method to degrade pollutants by using photo-induced electrons and photo-induced holes generated by semiconductor materials [19,20,21]. Many researchers have studied the process and mechanism of photocatalytic degradation of pollutants using PbS as a catalyst. Ajibade and Oluwalana synthesized nano-PbS particles and pointed out that they had excellent photocatalytic degradation effect on Rhodamine B [22]. Sehati and Entezari synthesized a PbS-CdS/Ti_6_O_13_ nano-photocatalyst and indicated its good photocatalytic degradation performance for dyes (Rb5) [23]. A PbS/MoS_2_ nanocomposite photocatalyst was fabricated by Raja et al., and its efficient photocatalytic degradation activity for methylene blue was proved [24]. The PbS quantum dots supported on polydopamine-encapsulated BiVO_4_ was gained through a self-assembled synthesis method and its good glyphosate photocatalytic degradation activity was verified [25]. Kar et al., synthesized a SnO_2_-PbS nanocomposite and suggested its excellent photocatalytic degradation performance for Rhodamine B under visible light irradiation [26].

In addition, as a typical 2D nanorod material, attapulgite has been widely used in composite photocatalysts. For example, Zhu et al., constructed an attapulgite intercalated mesoporous g-C_3_N_4_, which showed greatly enhanced photocatalytic activity for tetracycline degradation [27]. Zuo et al., synthesized an attapulgite/Cu_2_O/Cu/g-C_3_N_4_ compound material and displayed significant enhancement of photocatalytic activity for chloramphenicol degradation [28]. Integrated nanostructures of CeO_2_/attapulgite/g-C_3_N_4_ were synthesized by Ni and could be used as an efficient catalyst for photocatalytic desulfurization [29]. Hence, if the Pb element adsorbed on the attapulgite could be converted into PbS, that could not only provide excellent photocatalysis, but could also provide a new way for the comprehensive utilization of waste-adsorbed attapulgite.

Fortunately, according to our previous research, thiourea could vulcanize the metal elements (including Cd and Zn) adsorbed on the zeolite adsorbent under the hydrothermal reaction [30].
(1)$$\mathrm{zeolite}+\mathrm{Cd}/\mathrm{Zn}\overset{\mathrm{adsorption}}{\to }\mathrm{zeolite}-\mathrm{Cd}/\mathrm{Zn}$$
(2)$$\mathrm{SC}{\left({\mathrm{NH}}_{2}\right)}_{2}{+H}_{2}O\overset{\Delta }{\to }{\mathrm{CO}}_{2}{+2\mathrm{NH}}_{3}{+2H}^{+}{+S}^{2-}$$
(3)$$\mathrm{zeolite}-{\mathrm{Cd}/\mathrm{Zn}+S}^{2-}\overset{}{\to }\mathrm{zeolite}-\mathrm{CdS}/\mathrm{ZnS}$$

Thus, in theory, we should also be able to transfer the Pb element adsorbed on attapulgite to PbS through the vulcanization reaction by thiourea. Then the attapulgite with Pb adsorption saturation would be given excellent photocatalysis properties and could be applied in the photocatalytic degradation of pollutants.

The purpose of this research is to solve the thorny environmental problem of waste adsorbent, namely the Pb-adsorbed saturated waste attapulgite, and develop a novel photocatalyst for degrading pollutants. Based on the above analysis, in this research, the Pb-adsorbed saturated attapulgite (AT@Pb) was synthetized through mixing the simulated Pb wastewater with original attapulgite for enough time to reach the adsorption equilibrium. Then the AT@Pb was transferred to attapulgite/lead sulfide composites (AT@PbS) through the vulcanization reaction under hydrothermal synthesis conditions. Then, samples obtained were characterized by various advanced modern instruments to analyze the mineral phase composition, microscopic morphology, molecular functional groups, light absorption, electrochemical characteristics, free radical analysis, and so forth. Moreover, the AT@PbS was applied to explore the photocatalytic activities through the degradation of methylene blue (MB) under visible light. The findings confirmed that the AT@PbS photocatalyst displayed excellent photocatalytic activities, and provides a new approach for comprehensive utilization of waste adsorbents and cheap synthesis of a photocatalyst.

## 2. Experiment and Methods

### 2.1. Synthesis of Catalyst

The attapulgite (AT) used in this work was collected from Xuyi City, Jiangsu Province, China. It was milled with a ball mill to 1800 mesh for use. All the other chemical reagents (Analytical Reagent (AR); its content is greater than 99.5%, and its impurity content should be ignored) were purchased from China National pharmaceutical reagent company (Shanghai, China).

The waste adsorption-saturated attapulgite was obtained by static adsorption experiment. The specific experimental steps were as follows: 2 g of the above-described attapulgite were added into 500 mL of a PbCl_2_ solution (the concentration of Pb^2+^ was 500 mg/L), and then the mixture was continuously mixed for 6 h until its adsorption equilibrium was reached. After the reaction was completed, the mixture was put into a centrifuge tube, and centrifuged with deionized water and alcohol for several times. The mixture was filtered with a 0.45 μm filter membrane, and the filtered solid product was washed with a certain amount of alcohol. Finally, the filtered solid was dried at 75 °C to constant weight, and the waste adsorption-saturated attapulgite (AT@Pb) was obtained. 

As Figure 1 shows, the synthesis process of AT@PbS was as follows: the above-synthesized AT@Pb was added into 200 mL of deionized water and was magnetically stirred for 20 min. Then, 5 g of thiourea was added to the mixture and stirred for another 1 h. Thirdly, the above reactants were transferred to three 100 mL stainless steel autoclaves lined with Teflon, and reacted at 160 °C for 6 h in a blast drying oven. After the reaction was completed, the mixture was cooled naturally and the supernatant was poured out, then the remaining white powder was put into a centrifuge tube and centrifuged with deionized water and alcohol several times. And then, the centrifuged material was taken out after the water and reaction product completely separated. The synthetic product was dried in a blast drying oven at 105 °C to constant weight, and AT@PbS was acquired.

### 2.2. Materials Characterization

To clarify various properties of prepared materials, photocatalysts were characterized with the use of numerous pieces of advanced equipment. The morphologies of materials were observed using an FEI Talos F200s (USA) transmission electron microscope, equipped with high-angle annular dark-field scanning transmission electron microscopy (HAADF-STEM) and an FEI Super-X EDS detector (USA). The mineral composition of the synthesized material was measured by XRD patterns using a Shimadzu XD-3A diffractometer (Japan), employing the radiation of Cu-Kα (λ = 1.54056 Å). In addition, the surface chemical compositions of the catalysts were examined with a PHI 5000 VersaProbe X-ray photoelectron spectrometer (USA). The optical properties of the photocatalysts were detected by UV-Vis spectroscopy using a PerkinElmer Ultraviolet spectrophotometer (Shanghai, China). Additionally, the active species in the photocatalytic system were analyzed by electron spin resonance using an electron paramagnetic resonance spectrometer (EMXmicro-6/1/P/L, Karlsruhe, Germany), and the DMPO (5,5-dimethyl-1-pyrroline *N*-oxide) was selected as the capturing agent to trap the free radicals of ^•^O_2_^−^ and ^•^OH. Electrochemical impedance spectroscopy was conducted and photocurrent were measured with a CHI 660E electrochemical workstation (Shanghai, China).

### 2.3. Photocatalytic Experiments

In experiments, methylene blue (MB) was used as a pollutant to investigate the degradation effect of a photocatalyst under the irradiation of visible light (300 W Xenon-lamp). In each test, a photocatalyst (0.010 g) was put into 30 mL of MB solution to maintain an initial concentration of 10 mg/L. All the photocatalytic experiments were conducted at room temperature (25 °C, controlled by an indoor air conditioner), and the reaction liquids were put into a transparent photocatalytic reaction vessel. The photocatalytic reaction time was from 0 to 40 min, and the visible light was turned on at 0 min. After the photocatalytic degradation experiment, an ultraviolet visible spectrophotometer was used to measure the remnant concentration of MB at 664 nm.

In our research, to illustrate the photocatalytic mechanism of this reaction, the active species produced in the photocatalytic reaction were analyzed, and different types of capturing agents were put into the mixtures to quench the reactive species. These scavengers included 0.037 g of EDTA-2Na, 0.0094 g of TBA (*tert*-butyl alcohol), 0.0108 g of BQ (1,4-benzoquinone), and 0.0142 g of CCl_4_. In addition, to evaluate the photocatalytic properties of the recycled photocatalyst, consecutive cycles were done to determine the photocatalytic efficiencies of MB, and the photocatalyst was reused four times. After each recycled cycle, a 0.45 μm filter membrane was used to filter the photocatalyst material from the reaction solution, and the recycled sample were then washed by deionized water several times. After that, the recycled product was dried to a constant weight at 75 °C in a blast drying oven for the next usage. 

## 3. Results and Discussion

### 3.1. XRD

Figure 2 shows the X-ray diffraction analysis results of AT, AT@Pb, and AT@PbS. As seen from Figure 2, the mineral phases of the original AT were composed of quartz and attapulgite. Also, the corresponding diffraction peaks of quartz (PDF. No 65-0466) located at 2θ 20.9° (1 0 0), 26.7° (1 0 1), 50.1° (1 1 2), and 59.9° (2 1 1). The corresponding diffraction peaks of attapulgite (PDF. No 31-0783) were located at 2θ 13.7° (2 0 0), 16.3° (1 3 0), 19.8° (0 4 0), 21.3° (3 1 0), 28.0° (4 0 0), 30.98° (3 3 1), and 34.52–35.26° (0 6 1, 1 0 2 and 1 6 1). For natural mineral materials like attapulgite, it is common to contain a certain amount of quartz minerals [12,13]. After the adsorption saturation of Pb (AT@Pb), the position of XRD peaks remained basically stable, and two new peaks at 2θ 24.7° and 34.2° could be found. It was speculated that the adsorption of Pb caused some changes in the crystal structure. Compared with AT and AT@Pb, in the XRD analysis results of AT@PbS, in addition to the peaks of attapulgite and quartz, there were strong peaks at 2θ 26.0° (1 1 1), 30.1° (2 0 0), 43.1° (2 2 0), 51.0° (3 1 1), 53.4° (2 2 2), 68.9° (3 3 1), and 30.1° (4 2 0). According to the standard crystal database, the new peaks of AT@PbS belonged to PbS (PDF. No 65-2935). This result suggested that, the PbS with photocatalytic activity appeared after the vulcanization reactions of AT@Pb sample.

### 3.2. UV-Vis Diffuse Reflectance Spectra

Figure 3 displays the UV-Vis diffuse reflectance analysis results of AT, AT@Pb, and AT@PbS. As depicted in Figure 3a, both the AT (blue) and AT@PbS (red) had low visible light absorption, indicating that they could not be used as visible light catalysts. Comparison of the AT and AT@Pb revealed that the light absorption efficiencies of AT@PbS (pink) were significantly enhanced. Especially in the visible light range from 400 nm to 780 nm, its absorption efficiency was much higher than that of AT and AT@Pb. Moreover, the data of UV-Vis could be used to calculate the band gap value through the Tauc equation. According to the Tauc equation, the intercept between the straight line portion (in the hv and (αhν)^2^ relation curve of Figure 3a) and the X-axis should be the band gap value of the sample [31]. The analysis results (Figure 3b) showed that, in the energy range (from about 1.6 eV to 3 eV) of visible light, no obvious straight-line portion could be observed in the curves of AT and AT@Pb. Hence, we inferred that the band gap values of AT and AT@Pb were at least higher than 3 eV. At the same time, it could also be observed from Figure 3a that the band gap value of AT@PbS was about 1.41 eV, which might give this material excellent visible-light absorption capability. In conclusion, the UV-dis analysis results proved that, the vulcanization reaction made AT@PbS material possess outstanding absorbable properties for visible light. 

### 3.3. XPS

Figure 4 shows the XPS analysis results of AT@PbS. From the wide-scan results of Figure 4a, it can be seen that the main XPS peaks of AT@PbS included O 1s (531.1 eV), Si 2s (153.1 eV), Si 2p (102.1 eV), Al 2s (119.1 eV), Al 2p (74.1 eV), Pb 4f (142 eV and 137 eV), Pb 5p (88.2 eV), S 2p (162.1 eV), Mg 1s (1304.1 eV), and C 1s (284 eV). Among the above elements, except for C (analysis added), Mg, Si, Al, and O were the constituent elements of attapulgite. Additionally, Si and O were the constituent elements of quartz, and Pb and S were the constituent elements of PbS. Further, more details could be obtained from the high-resolution XPS analysis (from Figure 4b–g). As depicted Figure 4b, the peak of Si 2p could be divided into two peaks at 101.9 eV and 102.6 eV. These two peaks should be attributed to the different existence states of Si in attapulgite and quartz. The same reason might lead to the peak of O 1s, which also could be divided into two peaks at 531.4 eV and 533.4 eV (Figure 4d). Additionally, the Al 2p (73.9 eV at Figure 4c), Si 2p (101.9 eV), O 1s (531.4 eV), and Mg 1s (1303.9 eV in Figure 4e) were often observed in the XPS analysis results of the attapulgite sample [32]. From Figure 4f, it could be found that, the XPS peak of Pb 4f could be divided into two peaks at 141.6 eV and 136.7 eV. According to the position and area ratio, it could be inferred that the two peaks were Pb 4f_2/5_ and Pb 4f_2/7_, and that indicated the existence of Pb^2+^. Similarly, the peak of S 2p (Figure 4g) could be split into S 2p_1/2_ at 163.1 eV and S 2p_3/2_ at 162.1 eV, which suggested the existence of S^2−^. Accordingly, the simultaneous occurrence of characteristic peaks of Pb^2+^ and S^2−^ indicated the existing of PbS in the sample of AT@PbS [26]. In summary, the XPS analysis results were consistent with the fact that the AT@PbS sample contained attapulgite, quartz, and PbS.

### 3.4. TEM

Figure 5 depicts the TEM analysis results of AT@PbS. As depicted in Figure 5a, AT@PbS contained much nanorod-like material. The length of these nanorods was about 500 nm to 600 nm, and the width was only about 20 nm. These were typical microstructures of the attapulgite, and the phenomenon was basically consistent with the analysis results of SEM. With these nanorod-shaped materials, there were also considerable nanosquare-like particles, and the side length of these nanosquare particles was about 80 nm to 100 nm. Figure 5b–e shows the morphology and HRTEM analysis results of the nanosquare particles. As seen from Figure 5d,e, crystal planes with 0.178 nm spacing can be clearly observed, which obviously belonged to the PbS (3 1 1) crystal plane. Such results indicate that the nanoparticles were PbS. Meanwhile, the crystal planes with 0.638 nm spacing can be observed from Figure 5c,f. The existence of attapulgite crystal plane (2 0 0) once again confirmed that the nanorod-like material was attapulgite.

Figure 6 describes the element distribution analysis results of AT@PbS. It can be seen from Figure 6 that the elements Si, Al, O, and Mg were strongly distributed in the nanorods region, indicating that the nanorods were mainly composed of Si, Al, O and Mg. Such element composition was in line with that of attapulgite. At the same time, we could observe that the elements Pb and S were mainly distributed in the region of nanosquare particles, which was obviously in agreement with the constituent elements of PbS.

Figure 7 depicts the element distribution results along the line in Figure 6b. It can be seen from Figure 7b that, except for the voids between nanorods, Si, Al, O, and, Mg all displayed rich distributions in the region, and this finding was consistent with the element composition of attapulgite. Meanwhile, the distribution intensities of both Si and O elements on the line were significantly higher than that of both Al and Mg. This phenomenon indicates that the nanorod material probably also contained some quartz minerals, which was in line with the analysis results of XRD in Figure 2. On the other hand, from the Pb and S linear distribution, it can be seen that both Pb and S always appear intermittently together. The distance between the two occurrences was basically the same as that of the square nanoparticles in the EDS line. These results once again proved that the mineral phase of square nanoparticles was PbS. In conclusion, TEM results indicate that, after the vulcanization reaction, many nanosquare PbS particles were formed, while the original nanorod materials could also be observed. The appearance of these nanosquare PbS particles made the AT@PbS sample possess excellent visible light absorption and photocatalytic properties.

### 3.5. Photocatalytic Performance

Figure 8 showed the MB degradation results by different photocatalytic degradation systems. As depicted from Figure 8a, compared with other degradation systems (the AT+light system and the AT@Pb+light system), the AT@PbS+light system displayed much higher degradation performance. The MB in the solution (the original concentration was 20 mg/L) could be almost completely degraded in 40 min. This result further verified that the surface vulcanization reaction gave the material obvious photocatalytic properties. In addition, a certain amount of MB could also be degraded in the AT@PbS+dark system (about 19.2%) and the AT@PbS+light system (about 8.1%) before turning on the light, which indicated that AT@PbS material had adsorption capacity for MB. Generally speaking, this adsorption capacity could promote the contact between pollutants and photocatalysts in a certain degree, and that was very helpful for photocatalytic degradation [33].

Figure 8b,c presents the first-order kinetic [34,35] fitting results of the photocatalytic process in different reaction systems, and the original data was obtained from Figure 8a. It was observed that the first-order kinetic equation displayed excellent fitting results for all degradation systems, and all the fitting R^2^ values were above 0.94. These findings indicated that the apparent degradation process of MB in the AT@PbS+light system conformed to the first-order kinetics law. From the results in Figure 8c, it can also be seen that the kinetic constant of the AT@PbS+light system (0.1154 min^−1^) was much higher than that of other systems (0.0021 min^−1^ for the light system, 0.0075 min^−1^ for the AT+light system, 0.0036 min^−1^ for the AT@Pb+light system, and 0.0040 min^−1^ for the AT@PbS+dark system). Moreover, the UV-visible absorption spectra of degraded MB solution (AT@PbS+light system) are showed in Figure 8d. The results in Figure 8d reveal that, as the photocatalytic reaction proceeded, the MB characteristic peak (at about 740 nm) gradually decreased until it almost disappeared. This phenomenon again proves that the MB was gradually degraded with the progress of photocatalytic reaction. 

To investigate the photocatalytic stability of AT@PbS, the photocatalyst was reused four times to test its properties. Figure 9a depicts the recycled photocatalytic degradation efficiencies of the AT@PbS catalyst. It can be observed from Figure 9a that, after four cycles, the residue rate of MB in 40 min was only increased to 0.0386, which verified that the AT@PbS maintained excellent photocatalytic stability. Figure 9b shows the XRD pattern of AT@PbS after the 4th run. It can be observed that there was no obvious change in the peak position between the XRD results of the original AT@PbS and AT@PbS after the 4th run. The results indicate that the main mineral phase had not changed during the four cycles. Also, the characteristic peak of PbS (2θ at 30.1° and 43.1°) was slightly decreased, which indicates that there might have been a small amount of PbS loss during the recycling process. 

### 3.6. The Photocatalytic Degradation Mechanism

The photo-electric properties of materials were very important for the study of the photocatalytic mechanism. Figure 10 shows the transient photocurrent responses, electrochemical impedance spectra, and Mott-Schottky curves of AT, AT@Pb, and AT@PbS. As depicted from Figure 10a, the photo current feedback of AT@PbS was much stronger than that of AT and AT@Pb. This finding suggests that AT@PbS could be excited by the visible light to produce a large number of carriers (electrons and holes), which was crucial for the improvement of its photocatalytic performance. In addition, from the analytical results of electrochemical impedance spectra (Figure 10b), the order of the curve radius was as follows: AT > AT@Pb > AT@PbS, indicating that the resistance of AT@PbS was much less than that of AT@Pb and AT. This finding was consistent with the fact that AT@PbS contained many more carriers. 

The Mott-Schottky curves are exhibited in Figure 10c. It can be seen that the curve of AT@PbS was obviously different from that of AT and AT@PbS. In the curve of AT@PbS, within the range from about 0.25 V to 0.42 V, we can observe the characteristic curve of a typical n-type semiconductor. Additionally, the intersection of the fitting line for this curve and the voltage axis was about 0.157 V. This finding means that the flat band potential of the material was 0.157 V. Because the flat band potential of the general n-type semiconductor was about 0.2 V, higher than the conduction band potential [36], the conduction band potential of the material was about −0.043 V. Combined with the previous UV-V’s diffuse reflectance result, it could be concluded that the energy band range of the AT@PbS was about −0.043 V to 1.367 V. Additionally, this energy band range was very suitable to be a photocatalyst for the degradation of organic wastewaters.

Figure 11a shows the effect of various capturing agents on MB photocatalysis. The results were applied to further reveal the mechanism of AT@PbS photocatalytic degradation of MB. It was generally believed that [37,38], EDTA-2Na was a hole-trapping agent, TBA was considered to be a hydroxyl radical trapping agent, CCl_4_ was believed to be an electron trapping agent, and BQ was deemed an oxygen radical trapping agent. It is obvious from the results in Figure 11a that the effects of EDTA-2Na and TBA on MB degradation efficiencies were much greater than those of BQ and CCl_4_. This result indicates that the series of holes degradation path (Equations (4) and (5)) in the photocatalytic reaction system might be the main degradation mechanism of MB, and it probably played the major role in the photocatalytic degradation of MB. In contrast, the addition of CCl_4_ and BQ had little impact on the degradation efficiencies of MB, and the impact of BQ was the lowest. These results imply that, in the photodegradation system of AT@PbS, the electronic series degradation path (Equations (6) and (7)) could only play an auxiliary role during the photodegradation process of MB.
(4)$$H_{2}{O/\mathrm{OH}}^{-}{+h}^{+}\to \cdot \mathrm{OH}$$
(5)$$\cdot {\mathrm{OH}/h}^{+}+\mathrm{MB}\to \mathrm{Degradation}-\mathrm{products}$$
(6)$$O_{2}{+e}^{-}\to \cdot O_{2}{}^{-}$$
(7)$$\cdot O_{2}{}^{-}{/e}^{-}+\mathrm{MB}\to \mathrm{Degradation}-\mathrm{products}$$

Figure 11b shows the ESR spectra of radical adducts trapped by DMPO under the irradiation of visible light for DMPO-^•^OH. It could be seen from Figure 11b that, under the AT visible light system, almost no hydroxyl radical was observed, which once again confirms that AT material had almost no catalytic performance in visible light. In addition, in the AT@PbS visible light system, there was an obvious hydroxyl radical (1:2:2:1) characteristic signal [39], that once again proved the AT@PbS material obtained the photocatalytic activity of visible light, and the hydroxyl radicals probably played a crucial role in the photocatalytic degradation of MB. Additionally, Figure 11c describes the ESR spectra of radical adducts trapped by DMPO under the irradiation of visible light for DMPO-^•^O_2_^−^. As seen from Figure 11c, in both AT and AT@PbS visible light systems, no oxygen radical signal could be found. These results indicate that oxygen radicals could not contribute to the photocatalytic degradation of MB, which was consistent with the results in Figure 11a. In conclusion, the photo-generated holes and hydroxyl radicals played major roles in the degradation of MB under AT@PbS photocatalysis system.

Based on the above analysis, it can be inferred that the main MB degradation mechanism under the AT@PbS photocatalytic system was as follows: firstly, in the liquid phase photocatalytic system, the MB molecule was adsorbed on the surface of the attapulgite nanorods. At the same time, under the excitation of visible light, some valence band electrons of nano-PbS were excited to the conduction band to form photo-generated free electrons, while leaving some photo-generated holes in the valence band. Because the energy band range of AT@PbS was about −0.043 V to 1.367 V, and the oxidation ability of the free holes was obviously stronger than the reduction ability of the free electrons, the MB adsorbed on the attapulgite nanorod surfaces was degraded under the strong reactivity of the free holes and their derived hydroxyl radicals. Then, the new MB molecule continued to be adsorbed on the attapulgite nanorod surfaces and the photodegradation process was repeated until the pollutant was completely degraded (as depicted in Figure 12).

## 4. Conclusions

Through the comprehensive utilization of Pb-adsorbed saturated attapulgite (hazardous waste), a composite material (AT@PbS) with photocatalytic activity was synthesized. After the vulcanization synthetic reaction, the attapulgite retained its original nanorod morphology and the adsorbed Pb could be converted into nanosquare PbS (80–100 nm). Thus, the AT@PbS material was endowed with better visible light absorption, excellent visible photocatalysis, and good photo-electric properties. Using AT@PbS as the photocatalyst, MB could be completely degraded within 30 min under visible light. After four rounds of recycling, the photocatalytic degradation efficiencies remained stable. In addition, the AT@PbS energy band range was about −0.043 V to 1.367 V. The findings of this photodegradation mechanism illustrated that the photo-generated holes and their derived hydroxyl radicals were the main active substances for the photocatalytic degradation of the pollutant. The results of this study can provide a new approach for the comprehensive utilization of waste adsorbents for heavy metals.

## Figures and Tables

**Figure 1 ijerph-19-10457-f001:**
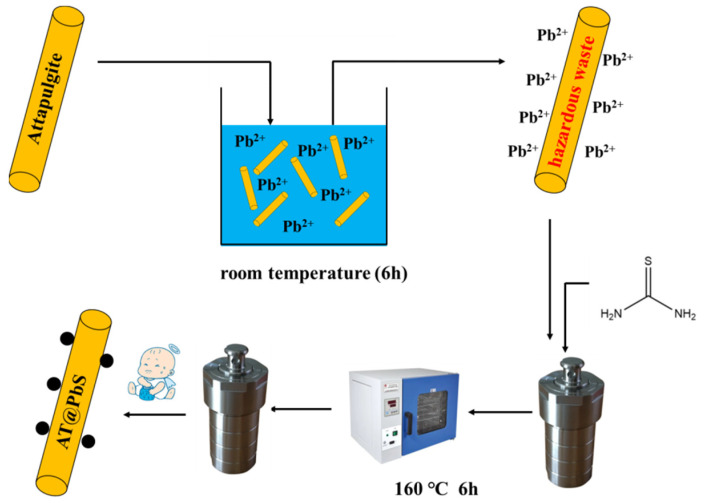
Schematic diagram of AT@PbS synthesis process.

**Figure 2 ijerph-19-10457-f002:**
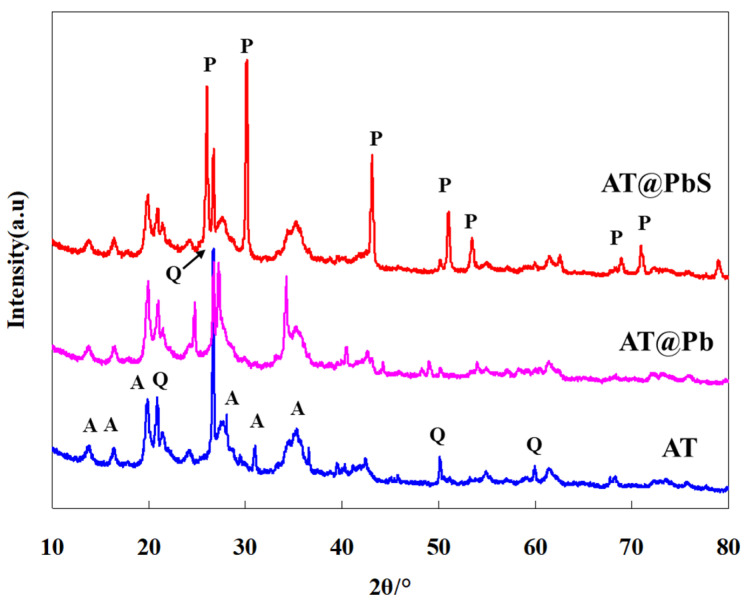
XRD (X-ray diffraction) patterns of AT (attapulgite), AT@Pb (Pb-adsorbed saturated attapulgite), and AT@PbS (attapulgite/lead sulphide composite).

**Figure 3 ijerph-19-10457-f003:**
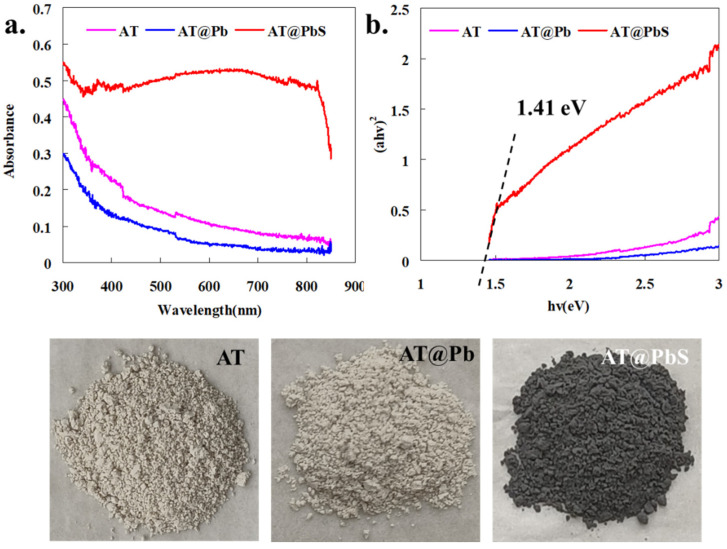
UV-Vis diffuse reflectance spectra for AT (attapulgite), AT@Pb (Pb-adsorbed saturated attapulgite), and AT@PbS (attapulgite/lead sulphide composite) (**a**), and the plotting results of band-gap energy for AT (attapulgit), AT@Pb (Pb-adsorbed saturated attapulgite), and AT@PbS (attapulgite/lead sulphide composite) (**b**).

**Figure 4 ijerph-19-10457-f004:**
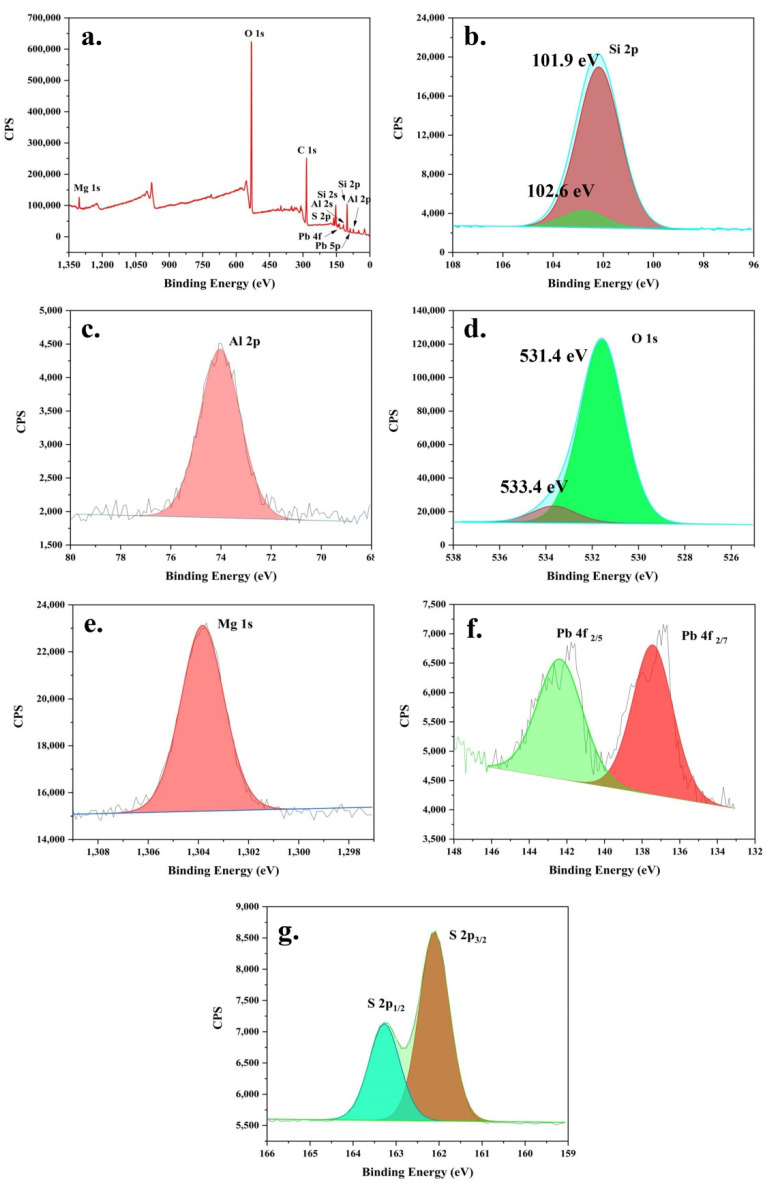
The wide-scan XPS (X-ray photoelectron spectrometer) of AT@PbS (attapulgite/lead sulphide composite) (**a**), the high-resolution XPS (X-ray photoelectron spectrometer) spectra of Si 2p (**b**), Al 2p (**c**), O 1s (**d**), Mg 1s (**e**), Pb 4f (**f**), and S 2p (**g**).

**Figure 5 ijerph-19-10457-f005:**
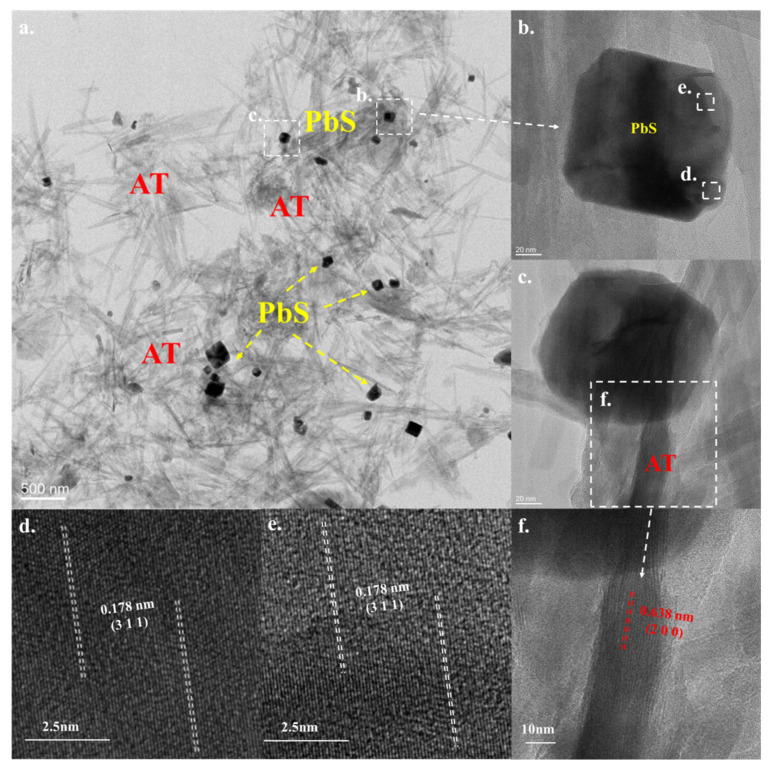
TEM (Transmission Electron Microscope) result (**a**–**c**) and HRTEM (High Resolution Transmission Electron Microscope) images (**d**–**f**) of AT@PbS (attapulgite/lead sulphide composite).

**Figure 6 ijerph-19-10457-f006:**
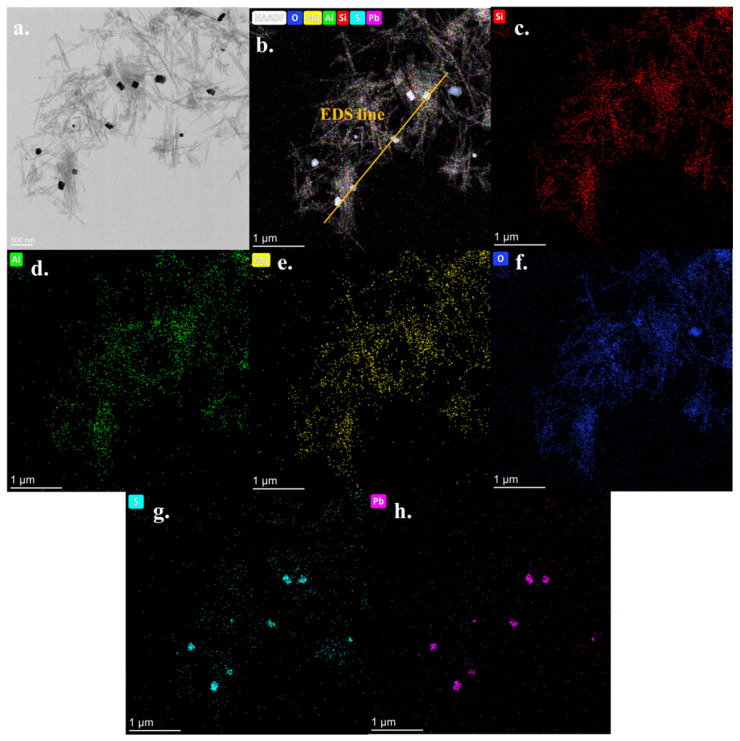
TEM (Transmission Electron Microscope)and HAADF (High-Angle Annular Field) results of AT@PbS (attapulgite/lead sulphide composite), element mapping results of (**a**) total elements, (**b**) Si, (**c**) Al, (**d**) Mg, (**e**) Mg, (**f**) O, (**g**) S, (**h**) Pb.

**Figure 7 ijerph-19-10457-f007:**
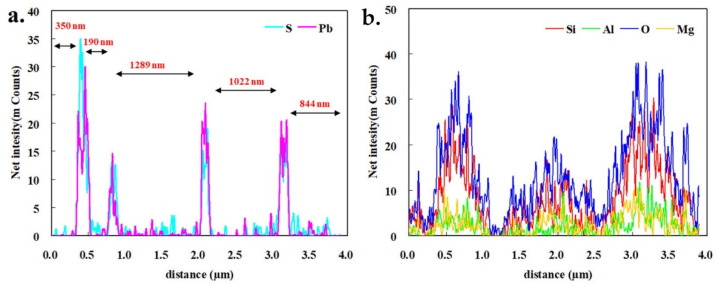
The element distribution results along line in Figure 6b ((**a**) S and Pb elements; (**b**): Si, Al, O and Mg elements).

**Figure 8 ijerph-19-10457-f008:**
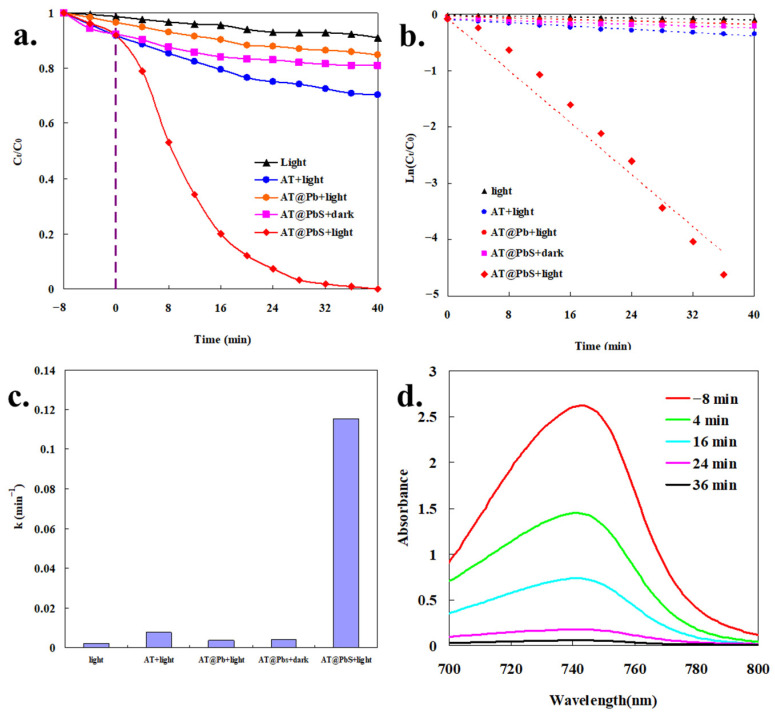
(**a**) The photocatalytic performance of different systems, (**b**) the apparent fitting findings using the first-order model, (**c**) the constants of first-order reactions for MB (methylene blue), (**d**) UV-visible absorption spectra of MB (methylene blue) solution degraded by AT@PbS+light (attapulgite/lead sulphide composite +light).

**Figure 9 ijerph-19-10457-f009:**
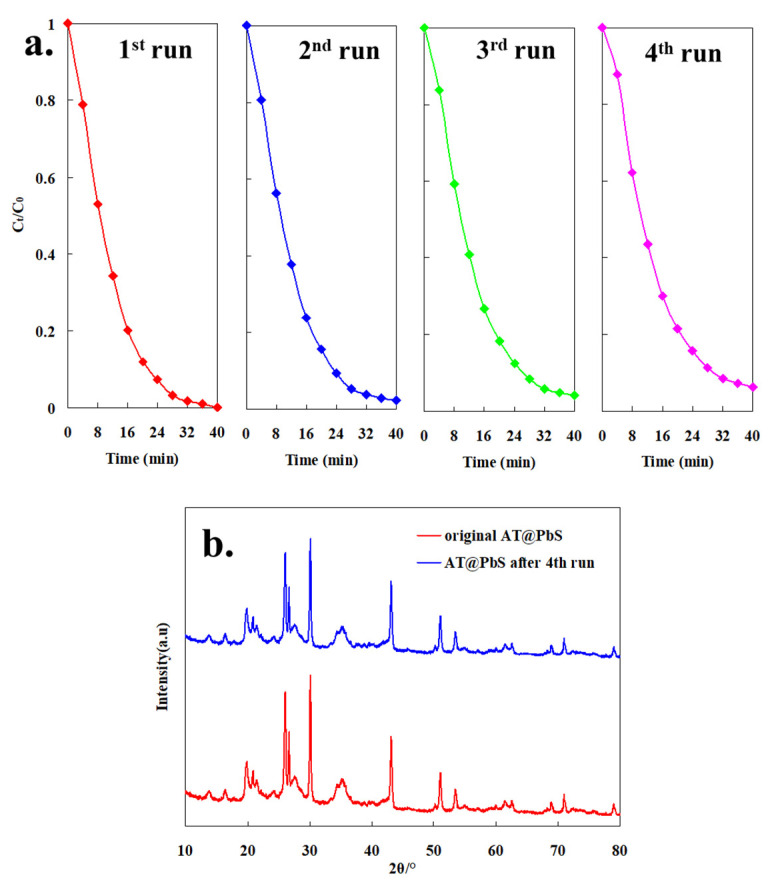
(**a**) The degradation curve of AT@PbS (attapulgite/lead sulphide composite) catalyst reused under visible light, (**b**) the XRD (X-ray diffraction) patterns of AT@PbS (attapulgite/lead sulphide composite) before and after 4th photocatalysis experiment.

**Figure 10 ijerph-19-10457-f010:**
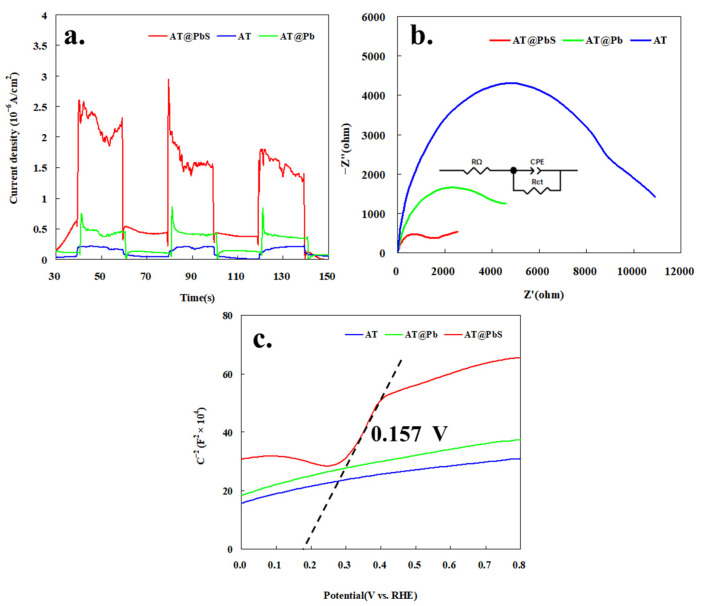
(**a**) Transient photocurrent responses of AT (attapulgite), AT@Pb (Pb-adsorbed saturated attapulgite), and AT@PbS (attapulgite/lead sulphide composite); (**b**) Electrochemical impedance spectra of AT (attapulgite), AT@Pb (Pb-adsorbed saturated attapulgite), and AT@PbS (attapulgite/lead sulphide composite), (**c**) Mott-Schottky curves of AT (attapulgite), AT@Pb (Pb-adsorbed saturated attapulgite), and AT@PbS (attapulgite/lead sulphide composite).

**Figure 11 ijerph-19-10457-f011:**
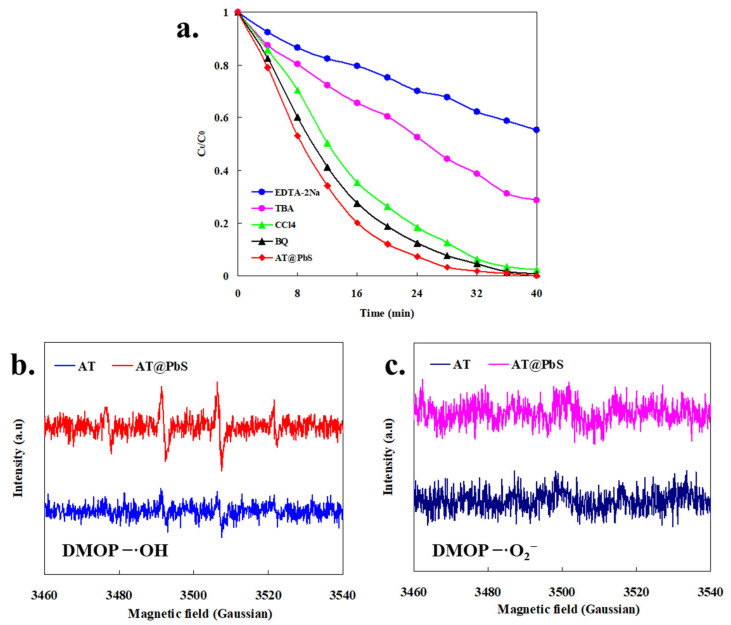
(**a**) The MB (methylene blue) photocatalytic efficiencies after the addition of capturing agents, (**b**) ESR (Electron Paramagnetic Resonance) spectra of radical adducts trapped by DMPO (5,5-dimethyl-1-pyrroline N-oxide) under the irradiation of visible light for DMPO-^•^OH (the combination of 5,5-dimethyl-1-pyrroline N-oxide and OH), (**c**) ESR (Electron Paramagnetic Resonance) spectra of radical adducts trapped by DMPO (5,5-dimethyl-1-pyrroline N-oxide) under the irradiation of visible light for DMPO-^•^O_2_^−^ (the combination of 5,5-dimethyl-1-pyrroline N-oxide and O_2_).

**Figure 12 ijerph-19-10457-f012:**
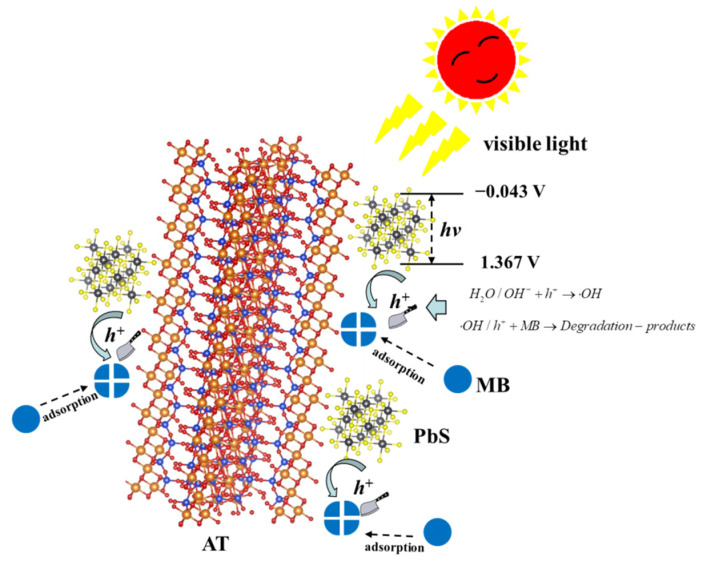
Schematic diagram of main degradation mechanism.

## Data Availability

Not applicable.

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
