# Peer review of "Making Pb Adsorption-Saturated Attapulgite with Excellent Photocatalysis Properties through a Vulcanization Reaction and Its Application for MB Wastewater Degradation"

_ijerph, 2022, doi:10.3390/ijerph191610457_

Round 1

Reviewer 1 Report

The article presents interesting research carried out in a comprehensive manner. The results of this study can provide a new approach for the comprehensive utilization of waste adsorbents for heavy metals.

However, I propose to supplement the article with a few elements.

Page 1 Line 37 - Give the chemical formula

Page 1 Line 41 - To explain the abbreviation

Page 2 Line 79-90 - Present the chemical reactions.

Page 3 Line 101 - What purity of the reagents were used? What kind of contaminants were possibly present in them?

Author Response

Thank you very much for the comment of review 1, the manuscript has been revised according to the comment, and the details of changes are as follows:

Page 1 Line 37 - Give the chemical formula

Reply: The chemical formula is provided in the Page 1 Line 37.

Page 1 Line 41 - To explain the abbreviation

Reply: According to the comment of reviewer, the abbreviation has been explained.

Page 2 Line 79-90 - Present the chemical reactions.

Reply: According to the comment of reviewer, the chemical reactions have been provided in the Page 2 Line 79-90.

Page 3 Line 101 - What purity of the reagents were used? What kind of contaminants were possibly present in them? 

Reply: All the other chemical reagents are analytical purity, and their contents are greater than 99.5%, and their impurity contents should be ignored.

Reviewer 2 Report

Article should be checked against plagiarism. There are sentences similar with other articles. overall similarity is low, however, introduction and sectin 2 to 3, there are sentences and phrases should be checked. 

Citations should be checked too.

There should be more detailed information about attapulgite. 

1. What is the main question addressed by the research?

Not-well-reported.

2. Do you consider the topic original or relevant in the field, and if
so, why?

Not original. Similar studies with other chemical have been done. It is a routine adsorption study.

3. What does it add to the subject area compared with other published
material?

Almost nothing.

4. What specific improvements could the authors consider regarding the
methodology?

Detailed characterization. Detailed study, detailed mathematical estimation. More parameters about the performance.  

5. Are the conclusions consistent with the evidence and arguments
presented and do they address the main question posed?

Yes.

6. Are the references appropriate?

Some reference from Zhang should be checked. Seems irrelevant.

7. Please include any additional comments on the tables and figures.

Well-done.

Author Response

Thank you very much for the comments, the manuscript has been revised according to the comment, and the details of changes are as follows: 

Article should be checked against plagiarism. There are sentences similar with other articles. Overall similarity is low, however, introduction and section 2 to 3, there are sentences and phrases should be checked. Citations should be checked too. There should be more detailed information about attapulgite. 

Reply: According to the comment of reviewer, the whole manuscript, including introduction, section 2 and section 3, has been checked for sentences and phrases, and they could be marked up using the “Track Changes” function. In addition, all citations have been checked, and more detailed information about attapulgite has been added in the manuscript.

1. What is the main question addressed by the research? 
Reply: The main question addressed by the research has been re-emphasized in the manuscript, and the revision has been marked up.

2. Do you consider the topic original or relevant in the field, and if so, why?
Reply: The topic is original and relevant in the field. The attapulgite, with well adsorption activity for Pb containing wastewater is a clay mineral. However, as a hazardous waste, the treatment of Pb adsorption saturated attapulgite was quite difficult. In this work, through simple vulcanization reaction, the waste Pb adsorption saturated attapulgite was transformed into composite materials (AT@PbS). AT@PbS displayed well visible light absorption, better photo-electric properties, excellent photodegradation performance for MB, and recycling stability. 

3. What does it add to the subject area compared with other published material? Reply: This work not only provides a new approach to construct the composite photocatalyst, but also demonstrates the possibility of the comprehensive utilization of waste heavy metal adsorbents. 

4. What specific improvements could the authors consider regarding the methodology?
Reply: Thank you very much for the comments of reviewer. We will pay more attention to the methodology. 

5. Are the references appropriate? 
Reply: All the references in the manuscript have been checked and revised, including reference from Zhang.